REGISTERED REPORT PROTOCOL

# Efficacy and safety of Dachaihu Decoction for acute pancreatitis: Protocol for a systematic review and meta-analysis

Xiang Xiao[1], Xuanyu Wu[1], Qinwei Fu[1], Xuelei Ren[1], Xiao Pang[2], Yuanyuan Li[2], Qinxiu Zhang[1]*, Yunhui Chen[3]*

1 Hospital of Chengdu University of Traditional Chinese Medicine, Chengdu University of Traditional Chinese Medicine, Chengdu, Sichuan, China, 2 Southwest Medical University, Luzhou, Sichuan, China, 3 College of Basic Medicine, Chengdu University of Traditional Chinese Medicine, Chengdu, Sichuan, China

☯ These authors contributed equally to this work.
* zhangqinxiu@cdutcm.edu.cn (QZ); chenyunhui@cdutcm.edu.cn (YC)

## Abstract

### Background

Dachaihu Decoction (DCD) is a traditional herbal formula widely used for treating acute pancreatitis (AP) in China. However, the efficacy and safety of DCD has never been validated, limiting its application. This study will assess the efficacy and safety of DCD for AP treatment.

### Methods

Relevant randomized controlled trials of DCD in treating AP will be searched through Cochrane Library, PubMed, Embase, Web of Science, Scopus, CINAHL, China National Knowledge Infrastructure database, Wanfang Database, VIP Database, and Chinese Biological Medicine Literature Service System database. Only studies published between the inception of the databases and May 31, 2023 shall be considered. Searches will also be performed in the WHO International Clinical Trials Registry Platform, Chinese Clinical Trial Registry, and ClinicalTrials.gov. Preprint databases and grey literature sources such as OpenGrey, British Library Inside, ProQuest Dissertations & Theses Global, and BIOSIS preview will also be searched for relevant resources. The primary outcomes to be assessed will include mortality rate, rate of surgical intervention, proportion of patients with severe acute pancreatitis transferred to ICU, gastrointestinal symptoms, and the acute physiology and chronic health evaluation II score. Secondary outcomes will include systemic complications, local complications, the normalization period of C-reactive protein, length of stay in the hospital, TNF-α, IL-1, IL-6, IL-8, and IL-10 levels, and adverse events. Study selection, data extraction, and assessment of bias risk will be conducted independently by two reviewers using the Endnote X9 and Microsoft Office Excel 2016 software. The risk of bias of included studies will be assessed by the Cochrane "risk of bias" tool. Data analysis will be performed using the RevMan software (V.5.3). Subgroup and sensitivity analysis will be performed where necessary.

**Data Availability Statement:** All relevant data are within the paper and its Supporting information files.

**Funding:** QXZ, Xinglin Scholars Scientific Research Promotion Plan of Chengdu University of Traditional Chinese Medicine-Innovation Team of Traditional Chinese Medicine Otorhinolaryngology Discipline, Natural Science (XKTD2021003); YHC, International Cooperation and Exchange Project of Sichuan Provincial Science and Technology Department (grant number 2017HH0004); YHC, the National Natural Science Foundation of China (grant number 81603537); YHC, the Project of Sichuan Provincial Administration of Traditional Chinese Medicine (grant number 2021MS464); YHC, the Youth Scholar Project of Chengdu University of Traditional Chinese Medicine (grant number QNXZ2019043). The funders had and will not have a role in study design, data collection and analysis, decision to publish, or preparation of the manuscript.

**Competing interests:** The authors have declared that no competing interests exist.

**Abbreviations:** AP, acute pancreatitis; APACHE II, acute physiology and chronic health evaluation II; CIs, confidence intervals; DCD, Dachaihu Decoction; MD, mean difference; MODS, multiple organ dysfunction syndrome; RCTs, randomized controlled trials; RR, risk ratio; SIRS, systemic inflammatory response syndrome; SMD, standard mean difference; TCM, traditional Chinese medicine.

# Results

This study will provide high-quality current evidence of DCD for treating AP.

# Conclusion

This systematic review will provide evidence of whether DCD is an effective and safe therapy for treating AP.

# Trial registration

**PROSPERO registration number** CRD42021245735. The protocol for this study was registered at PROSPERO, and is available in the S1 Appendix. https://www.crd.york.ac.uk/prospero/display_record.php?ID=CRD42021245735.

# Introduction

## Description of the condition

Acute pancreatitis (AP) is an inflammatory disorder of the pancreas characterized by edema, hemorrhage, and necrosis of the pancreas [1]. AP is caused by multiple factors, and it is one of the most common acute abdominal diseases, with an annual prevalence estimate of 40~45 per 100,000 individuals in the United States [2]. The typical clinical manifestations of AP include high serum amylase and lipase levels and digestive system symptoms such as acute abdominal pain, abdominal distention, nausea, and vomiting [3]. The mortality of AP is about 3 to 6% but may reach 10% to 30% in severe cases [4]. The common AP complications, include systemic inflammatory response syndrome (SIRS), multiple organ dysfunction syndrome (MODS), infectious pancreatic necrosis, septicemia, and abdominal compartment syndrome [5–7]. Despite the rapid development of diagnosis techniques, and management methods for AP, the mortality rate related to the disease is still high [8,9]. Furthermore, there no specific drug for AP treatment. Additionally, the use of antibiotics and surgical intervention remains controversial [10–12]. Hence, there is an urgent need to explore AP treatment options. Complementary therapies could also improve treatment outcomes.

## Description and working meachanism of DCD

Traditional Chinese medicine (TCM) is an alternative and complementary medicine widely used in treating various diseases. Dachaihu Decoction (DCD) is a classical TCM from "*Treatise on Febrile Disease (Shanghan Lun)*" first reported by Zhongjing Zhang (150–219 A.D.). It is a concoction of TCM consisting of Rhei Radix et Rhizoma (dahuang, 1g), Bupleuri Radix (chaihu, 6g), Aurantii Fructus Immaturus (zhishi, 2g), Scutellariae Radix (huangqin, 3g), Pinelliae Rhizoma (banxia, 4g), Paeoniae Radix Alba (shaoyao, 3g), Jujubae Fructus (dazao, 3g), and Zingiberis Rhizoma Recens (shengjiang, 1g) [13]. The efficacy and safety of DCD in treating angina, acute cholecystitis, type 2 diabetes, fatty liver disease, and AP have been reported in numerous clinical trials [14–18]. DCD in combination with ulinastatin and somatostatin has recently been used to treat severe AP. DCD reduced the IL-8, CRP, and TNF-α level, and reduced the APACHE II score after just one week without noticeable side effects [19]. DCD in combination with ulinastatin relieved gastrointestinal symptoms, reduced the serum amylase, TNF-α, and IL-6 levels in patients with AP [20]. In addition, DCD and dexamethasone therapy

for one week improved the pancreatic blood flow (blood flow, blood volume, and permeability surface) and restored the normal level of gastrointestinal hormones (vasoactive intestinal peptide, gastric inhibitory polypeptide, pancreatic polypeptide, cholecystokinin, gastrin, and the recovery time of abdominal pain, abdominal distention, bowel sound, and exhaust) in patients with severe AP, relative to patients that received dexamethasone alone. No side effects were observed [21]. The above clinical trials provided evidence that DCD could be effective in treating AP.

$Ca^{2+}$ overload and trypsinogen activation play a key role in the pathogenesis mechanisms of AP [22]. Available literature has indicated that DCD relieves AP treatment through multiple pathways and targets. Studies have demonstrated that baicalin in DCD modulates inflammation by suppressing the expressions of TNF, IL-1, IL-6, PKD1, NF-κB, and P-selectin protein. DCD also suppresses apoptosis by upregulating the expression of caspase-3 and down-regulating that of Bax protein [23–26]. Emodin in DCD protects against intestinal mucosal barrier injury by increasing the expression of miR-218a-5p and suppresses inflammation by modulating the expression of serum amylase, lipase, MPO, VDAC1, NLRP3, VDAC1, TNF-α, and IL-18 and modulating the P2X7/NLRP3 signaling pathway [27–29]. Besides, some studies have indicated that Emodin prevents acute lung injury caused by severe AP by inhibiting neutrophil proteases activity, and regulating the expression of NLRP3 and CIRP, and the expression of long non-coding RNA-mRNA networks [30–33]. Modified DCD inhibit inflammatory reactions of the pancreas, ileum, and lung by regulating the expression of occludin and NF-κB [34]. DCD could decrease intracellular $Ca^{2+}$ concentration and inhibit the $Ca^{2+}$-$Mg^{2+}$ ATPase activity [35]. DCD alleviates AP by regulating inflammation and inhibiting apoptosis.

## The significance of this review

Currently, the efficacy of DCD in treating AP remains controversial. DCD has shown potential as an effective, safe, and promising therapy to treat AP. Although many clinical trials on the efficacy and safety of DCD in treating AP have been reported, high-quality evidence on these concerns is lacking. In addition, DCD therapy has been linked to certain adverse events, such as dizziness, peritonitis, bleeding, pancreatic abscess, nausea, and vomiting [36–38]. Consequently, it is critical to assess the efficacy and safety of DCD therapy based on a systematic review and meta-analysis, which provides summarizes and provides very high medical evidence [39]. To the best of our knowledge, this is the first systematic review and meta-analysis to assess the efficacy and safety of DCD for AP treatment based on RCTs. The findings of this study will provide research-backed evidence on the application of DCD for AP treatment.

## Objectives

This systematic review aims to assess the efficacy and safety of DCD in combination with or without conventional therapy in managing AP.

## Methods

### Study registration and design

This systematic review was registered on PROSPERO online platform (https://www.crd.york.ac.uk/prospero/) under the registration number CRD42021245735. This protocol was reported following the Preferred Reporting Items for Systematic Review and Meta-Analysis Protocols (PRISMA-P) guidelines [40]. The PRISMA-P checklist to be used is shown in S2 Appendix. This systematic review and meta-analysis will be performed and reported following the PRISMA harms checklist [41].

## Inclusion criteria of the relevant studies

**Types of studies.** Only RCTs evaluating the effectiveness and safety of DCD for AP treatment will be considered. There will be no restriction on the year of publication, but only studies published in Chinese and English will be considered. To increase the data scope, we consider incorporating grey literature and preprint servers. Non-RCTs, case reports, commentary, literature review, and protocol will be excluded.

**Types of participants.** Different subtypes of AP, including biliary AP, alcoholic AP, hypertriglyceridemic AP, mixed causes AP, and other/idiopathic AP will be included. All AP cases, regardless of their age, gender and ethnicity, or initial severity will be included. AP diagnosis will be based on the guidelines for the diagnosis and treatment of acute pancreatitis (2014), the Guidelines for diagnosis and treatment of acute pancreatitis in China (2013, Shanghai), and the Guidelines for the management of acute pancreatitis developed at the Bangkok conference [42–44]. Thus, AP will be considered when 2 of the 3 following criteria are fulfilled: (1) Abdominal pain consistent with AP; (2) the serum amylase and/or lipase concentration at least 3 times higher than the upper limit of the normal level; (3) abdominal imaging results in line with AP imaging changes.

**Types of interventions.** Participants in the experimental group should have received DCD alone or combined with conventional treatments. The administration route, dosage and course of treatment of the intervention group and the control group were consistent. Studies on other TCM therapies, such as acupuncture, cupping, foot baths, and other Chinese herbal medicines will be excluded.

**Group comparisons.** Participants in the control group should have received conventional therapies, including fluid resuscitation, antibiotic therapy, nutritional support, and mechanical ventilation among others [7].

**Outcome measurements.** The primary outcomes will include mortality rates, rate of surgical intervention, proportion of patients with severe AP transferred to ICU, gastrointestinal symptoms (relief time of abdominal pain and bloating, recovery time of anal exhaust, defecation, and bowel sound), and the acute physiology and chronic health evaluation (APACHE) II score, which indicates the prognosis and severity of AP [45]. The secondary outcomes will include systemic complications (SIRS, septicemia, MODS, and abdominal compartment syndrome), local complications, the serum CRP, length of stay in hospital, TNF-α counts, IL-1 counts, IL-6 counts, IL-8 counts, IL-10 counts, and other adverse events.

## Search and extrcation of the relevant studies

**Electronic searches.** Relevant RCTs will be searched through Cochrane Library, PubMed, Embase, Web of Science, Scopus, CINAHL, the China National Knowledge Infrastructure (CNKI) database, the Wanfang Database, the VIP Database, and the Chinese Biological Medicine Literature Service System (SinoMed) databases. Only RCTs published between the inception of the databases and May 31, 2023. The search was performed independently by two researchers. The detailed search strategy for PubMed is shown in Table 1, and the detailed search strategy for other databases is available in S3 Appendix.

**Other search resources.** Relevant unpublished RCTs in the WHO International Clinical Trials Registry Platform (https://www.who.int/ictrp/en/), Chinese Clinical Trial Registry (http://www.chictr.org/cn/) and ClinicalTrials.gov (http://clinicaltrials.gov) will also be included. Preprint servers and grey literature sources such as OpenGrey, British Library Inside, ProQuest Dissertations & Theses Global, and BIOSIS preview will also be included. In addition, RCTs in the reference lists of relevant studies will also be included.

**Table 1. Search strategy for PubMed.**

| NO. | Search terms |
|-----|--------------|
| #1 | Pancreatitis |
| #2 | Acute Pancreatitis |
| #3 | Acute Edematous Pancreatitis |
| #4 | Peripancreatic Fat Necroses |
| #5 | Pancreatitis, Acute Necrotizing |
| #6 | Hemorrhagic Necrotic Pancreatitis |
| #7 | Acute Necrotizing Pancreatitis |
| #8 | OR/#1-#7 |
| #9 | Dachaihutang |
| #10 | Da-Chai-Hu-Tang |
| #11 | Dachaihu Decoction |
| #12 | OR/#9-#11 |
| #13 | #8 AND #12 |
| #14 | randomized controlled trial |
| #15 | controlled clinical trial |
| #16 | randomized |
| #17 | Randomly |
| #18 | Trials |
| #19 | RCT |
| #20 | OR/#14-#19 |
| #21 | #13 AND #20 |

## Data collection and analysis

**Selection of studies.** Relevant studies will be selected by two reviewers independently using the EndnoteX9 software. Duplicate documents will be eliminated through electronic and manual-based steps. The titles and abstracts will be screened to exclude literature that does not meet the inclusion criteria. Studies deemed suitable will be further screened through full-text reading. The results will be cross-checked to ensure consistency. A third reviewer will be consulted if discrepancies will be observed in the results. The entire selection process will be performed in line with the PRISMA-style as shown by the flow chart in S1 Fig.

**Data extraction and management.** Data extraction and management will be conducted by two reviewers independently into a standard excel spreadsheet. Important information such as the name of first author, year of publication, country of publication, information of participant (age, duration and severity of AP, etc.), sample size, intervention group and control group (administration route, dosage, and duration), and primary and secondary outcomes at all reported time points will be collected. The results will be cross-checked by two reviewers. Any discrepancies will be resolved by a third reviewer.

**Assessment of risk of bias.** The risk of bias of included studies will be determined by two reviewers using the Cochrane risk-of-bias tool in terms of the following seven aspects: random sequence generation, allocation concealment, blinding of participants and personnel, blinding of outcome assessment, incomplete outcome data, selective reporting, and other bias [46]. The risk of bias for each study will be graded as high, low, or unclear. Any disagreements will be resolved through discussion.

**Measures of treatment effect.** Mean difference (MD) or standard mean difference (SMD) with 95% confidence intervals (CIs) will be employed to analyze continuous data. Risk

ratio (RR) or odds ratio with 95% CIs will be calculated for dichotomous data. Statistically significance will be set at $p<0.05$.

**Dealing with missing data.** For studies without clear or complete data, the corresponding author will be contacted by e-mail or telephone. If accurate data are still unavailable, the study will be excluded. The impact of missing data will be assessed through sensitivity analysis.

**Assessment of heterogeneity.** All statistical analyses will be performed using the RevMan software (V.5.3). Heterogeneity will be assessed using the inconsistency index ($I^2$) and Cochrane's Q test. Heterogeneity will be graded as high ($I^2 > 75\%$), moderate (between 50% and 75%), or low ($I^2 < 50\%$). For Cochrane's Q test, $p<0.05$ indicates significant heterogeneity. Subgroup analysis or narrative analysis will be conducted if high heterogeneity is observed among studies.

**Data synthesis.** According to the Cochrane guideline, the fixed-effects model will be used to combine the effect size when heterogeneity is low, the random-effects model will be employed to combine the effect size if heterogeneity is moderate, and subgroup analysis or meta-regression will be performed to interpret significant heterogeneity where feasible. A narrative analysis of the results will be conducted if the meta-analysis is unfeasible.

**Subgroup analysis.** If heterogeneity is significant, subgroup analyses will be conducted. Similarly, subgroup analysis will be performed in terms of the severity of AP, the routes of administration, age, and gender.

**Sensitivity analysis.** Sensitivity analysis will be carried out to evaluate the robustness of treatment effects of included studies based on the high risk of bias, missing data, or sample size. This test will attempt to determine whether a single study accounted for the significant heterogeneity.

**Publication bias.** Standard funnel plots and Egger's regression test will be utilized to explore publication bias in meta-analyses with more than 10 studies.

## Patient and public involvement statement

No patients or public will be involved.

## Ethics and dissemination

Ethical approval is not applicable since this is a protocol for systematic review and meta-analysis that does not include actual patients or require data privacy. The finding of this study will be published in a peer-reviewed journal, the PROSPERO website, and relevant academic conferences presentation.

## Discussion

Clinically, AP is an acute inflammatory disease of the pancreas with severe complications and high mortality. It has been shown that DCD has been used for the treatment of AP for thousands of years in China. This systematic review and meta-analysis is designed to assess the efficacy and safety of DCD in AP treatment. The results of this study are expected to inform formulation of evidence-based guidelines for AP treatment. If any changes are to be made in protocol, we will provide the date, change, and reason of each amendment.

## Supporting information

**S1 Fig. The flow diagram of the study selection process.**
(PDF)

**S1 Appendix. The protocol registed on PROSPERO.**
(PDF)

**S2 Appendix. The PRISMA-P checklist.**
(PDF)

**S3 Appendix. The search strategy for all the databases.**
(PDF)

## Acknowledgments

We thank home for researchers(https://www.home-for-researchers.com) for the language polishing.

## Author Contributions

**Conceptualization:** Xiang Xiao, Qinxiu Zhang, Yunhui Chen.

**Data curation:** Xiang Xiao, Xuanyu Wu, Qinwei Fu.

**Formal analysis:** Xuelei Ren, Xiao Pang, Yuanyuan Li.

**Funding acquisition:** Qinxiu Zhang, Yunhui Chen.

**Investigation:** Xiang Xiao, Qinwei Fu, Yunhui Chen.

**Methodology:** Xiang Xiao, Xuanyu Wu, Qinxiu Zhang, Yunhui Chen.

**Project administration:** Qinxiu Zhang, Yunhui Chen.

**Supervision:** Xuanyu Wu, Qinxiu Zhang, Yunhui Chen.

**Validation:** Qinwei Fu, Xuelei Ren, Xiao Pang, Yuanyuan Li.

**Writing – original draft:** Xiang Xiao, Xuanyu Wu, Yunhui Chen.

**Writing – review & editing:** Xiang Xiao, Xuanyu Wu, Yunhui Chen.

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
