## [Decision Letter · Decision Letter 0]

24 Nov 2022

PONE-D-21-34892Efficacy and safety of Dachaihu Decoction for acute pancreatitis: protocol for a systematic review and meta-analysisPLOS ONE

Dear Dr. Xiao,

Thank you for submitting your manuscript to PLOS ONE. After careful consideration, we feel that it has merit but does not fully meet PLOS ONE’s publication criteria as it currently stands. Therefore, we invite you to submit a revised version of the manuscript that addresses the points raised during the review process.

ACADEMIC EDITOR: Please check the comments of all reviewers and avoid excessive acronyms in the abstract and throughout the manuscript.

We look forward to receiving your revised manuscript.

Kind regards,

Marcus Tolentino Silva

Academic Editor

PLOS ONE

Journal Requirements:

Reviewers' comments:

Reviewer's Responses to Questions

**Comments to the Author**

1. Does the manuscript provide a valid rationale for the proposed study, with clearly identified and justified research questions?

Reviewer #1: Yes

Reviewer #2: Partly

2. Is the protocol technically sound and planned in a manner that will lead to a meaningful outcome and allow testing the stated hypotheses?

Reviewer #1: Yes

Reviewer #2: Partly

3. Is the methodology feasible and described in sufficient detail to allow the work to be replicable?

Reviewer #1: Yes

Reviewer #2: Yes

4. Have the authors described where all data underlying the findings will be made available when the study is complete?

Reviewer #1: Yes

Reviewer #2: Yes

5. Is the manuscript presented in an intelligible fashion and written in standard English?

Reviewer #1: Yes

Reviewer #2: No

6. Review Comments to the Author

You may also provide optional suggestions and comments to authors that they might find helpful in planning their study.

Reviewer #1: This protocol for systematic review aimed to assess the efficacy and safety of Dachaihu Decoction (DCD) for the treatment of acute pancreatitis (AP). This protocol has been well described and written, and has been registered on PROSEPEPRO. However, I have several concerns and questions. The detailed comments are listed below.

(1) Line 16: Please provide all the ORCIDs of the authors, if possible.

(2) Line 28-30: If the DCD has been widely administered as an effective treatment, why did a systematic review require? Please point out why this systematic review is necessary, such as there is no sound evidence for this treatment, etc.

(3) Line 32-37, 151: Please also consider electronic datasets (Scopus, CINAHL), grey literature sources (OpenGrey, British Library Inside, ProQuest Dissertations & Theses Global, and BIOSIS preview), and preprint servers, for literature search.

(4) Line 55: Please also provide the PDF of your protocol on PROSEPERO as Supplementary Material.

(5) Line 86-86: The DCD has been well described. Please consider provide the amount of these medicinal materials used in the DCD.

(6) Line 129: Maybe “has been proved”  “has shown potential”? The aim of this review is to provide high-quality evidence to “prove” it.

(7) Line 133: More description on why the safety of DCD needed to be systematically reviewed. Please provide some disadvantage or adverse effects of DCD in AP treatment to show why this systematic review is needed, maybe the safety of DCD has not been determined?

(8) Line 153-158: Please provide the specific diagnostic criteria of AP for inclusion instead of citing references.

(9) Line 178: Chinese Science Journal Database (VIP)?

(10) Line 219-222: What is the criteria for Cochrane’s Q statistics?

(11) Line 208-209, 223-234: Please provide the plan for dealing with publication bias in a separated section.

(12) Figure S1: The PRISMA 2020 checklist has been published. Please applied the diagram in this latest guideline.

Reviewer #2: The authors of this study have developed a protocol for meta-analysis to assess the efficacy and safety of Dachaihu Decoction for acute pancreatitis

There are a few concerns abouth this protocol

Types participants: The authors have to specify the etiology of pancreatitis to assess the efficacy. They have to include studies that have administered this decoction either to alcoholic pancreatitis or to idiopathic pancreatitis patients. They cannot generalize the assessment.

Types of interventions: The authors need to include studies that have administered a uniform dosage. Otherwise there may be a hetergenity as the authors have mentioned in the discussion. Since this is only a protocol, they can modify to include uniform dosage, route of administration and duration/severity of disease.

All other methods are acceptable

English is poor, lot of confusion in the subheading types of comparisons. needs to improve.

7. PLOS authors have the option to publish the peer review history of their article (what does this mean?). If published, this will include your full peer review and any attached files.

Reviewer #1: No

Reviewer #2: No

---

## [Author Response · Author response to Decision Letter 0]

19 Dec 2022

Dear reviewers:

We are very grateful for your efforts to deal with our manuscript, and also very grateful for the professional and pertinent comments you have put forward. We have revised the corresponding part according to the suggestions of the two reviewers. In this reply, we responded to the comments one by one.

Reviewer #1: This protocol for systematic review aimed to assess the efficacy and safety of Dachaihu Decoction (DCD) for the treatment of acute pancreatitis (AP). This protocol has been well described and written, and has been registered on PROSPEPRO. However, I have several concerns and questions. The detailed comments are listed below.

(1) Line 16: Please provide all the ORCIDs of the authors, if possible.

We have added the ORCID IDs of all authors as requested (lines 14-21).

(2) Line 28-30: If the DCD has been widely administered as an effective treatment, why did a systematic review require? Please point out why this systematic review is necessary, such as there is no sound evidence for this treatment, etc.

As we reported in the manuscript, DCD was widely used in the treatment of AP, but its safety and effectiveness have not been verified, which limits its application. So we designed this systematic review and meta-analysis (lines 28-30).

(3) Line 32-37, 151: Please also consider electronic datasets (Scopus, CINAHL), grey literature sources (OpenGrey, British Library Inside, ProQuest Dissertations & Theses Global, and BIOSIS preview), and preprint servers, for literature search.

Thank the reviewer for the comment on the literature retrieval. We agree with the reviewer's suggestion and take electronic databases including Scopus and CINAHL, grey literature sources including OpenGrey, British Library Inside, ProQuest Dissertations&Themes Global, and BIOSIS preview, and preprint servers as our data sources (lines 32-40, 180, 191-193).

(4) Line 55: Please also provide the PDF of your protocol on PROSEPERO as Supplementary Material.

We have uploaded the protocol registered on the PROSPERO as a supplementary material (line 57-58).

(5) Line 86-86: The DCD has been well described. Please consider provide the amount of these medicinal materials used in the DCD.

Thanks for the approval and proposal of the reviewer, and we have reported the amount of the medicinal materials used in DCD (lines 89-92).

(6) Line 129: Maybe “has been proved”  “has shown potential”? The aim of this review is to provide high-quality evidence to “prove” it.

We are sorry for such a mistake in the manuscript, and we have corrected it to make sure the correctness of the manuscript (lines 124-125).

(7) Line 133: More description on why the safety of DCD needed to be systematically reviewed. Please provide some disadvantage or adverse effects of DCD in AP treatment to show why this systematic review is needed, maybe the safety of DCD has not been determined?

Some studies suggested that when DCD was used to treat AP, there might be adverse reactions such as dizziness, peritonitis, bleeding, pancreatic abscess, nausea, and vomiting. We believe that the possible adverse reactions can not be ignored while evaluating its efficacy. Therefore, this study will also explore the safety of DCD (lines 127-129).

(8) Line 153-158: Please provide the specific diagnostic criteria of AP for inclusion instead of citing references.

We have provided the specific diagnostic criteria of AP in the manuscript according to the requirements of the reviewer (lines 157-160).

(9) Line 178: Chinese Science Journal Database (VIP)?

I'm sorry that such an inappropriate abbreviation appears in the manuscript. The correct database name is "VIP Database", which has been added to the text (lines 34; lines 181).

(10) Line 219-222: What is the criteria for Cochrane’s Q statistics?

Heterogeneity will be assessed using the inconsistency index (I2) and Cochrane's Q test. Heterogeneity will be graded as high (I2 > 75%), moderate (between 50% and 75%), or low (I2 < 50%). And the statistical criteria for Cochrane’s Q test are as follows: when P<0.05, there is significant heterogeneity, otherwise, there is no heterogeneity. We have added the above to the manuscript (lines 227-228).

(11) Line 208-209, 223-234: Please provide the plan for dealing with publication bias in a separated section.

We have described “Publication Bias” as an independent chapter according to the requirements of the reviewer (lines 243-244).

(12) Figure S1: The PRISMA 2020 checklist has been published. Please applied the diagram in this latest guideline.

Thanks for the proposal of the reviewer, we have designed the study selection process according to PRISMA 2020 Flow Diagram and uploaded it as a supplementary material (lines 202, 269).

Reviewer #2: The authors of this study have developed a protocol for meta-analysis to assess the efficacy and safety of Dachaihu Decoction for acute pancreatitis. There are a few concerns abouth this protocol

(1) Types participants: The authors have to specify the etiology of pancreatitis to assess the efficacy. They have to include studies that have administered this decoction either to alcoholic pancreatitis or to idiopathic pancreatitis patients. They cannot generalize the assessment.

We have specified the disease type according to the reviewer's opinion. Any type of AP, including biliary AP, alcoholic AP, hyperlipidemic AP, mixed AP, and other/idiopathic AP, will be included in our study (lines 151-152). This means that we have not limited the etiology of AP, because the clinical symptoms of AP caused by various causes are similar, and the TCM treatment is also the same.

(2)Types of interventions: The authors need to include studies that have administered a uniform dosage. Otherwise there may be a hetergenity as the authors have mentioned in the discussion. Since this is only a protocol, they can modify to include uniform dosage, route of administration and duration/severity of disease.

Thanks for the comments of the reviewers, we ignored the consistency of administration route, drug dose, course, and severity of AP in the study. According to the comment of the reviewers, we improved the type of intervention: the administration route, dosage, and course of treatment of the intervention group and the control group were consistent. However, we do not require all the included studies to use the same route of administration, drug dose, and course of treatment, because each TCM researcher may have different clinical thinking and administration habits. If the above causes lead to high heterogeneity between studies, we will conduct a subgroup analysis. (line 162-163).

(3)English is poor, lot of confusion in the subheading types of comparisons. needs to improve.

In order to improve the quality and readability of the manuscript, we modified the entire text and invited native speakers to modify the sentence and grammar of the manuscript. At the same time, we have submitted the certificate of English editing as a supplementary document. In addition, we have appropriately revised some subheadings in the manuscript (line 86, 123, 145, 166, 169, 178, 188). 

Yours sincerely,

Xiang Xiao

---

## [Decision Letter · Decision Letter 1]

24 Jan 2023

PONE-D-21-34892R1Efficacy and safety of Dachaihu Decoction for acute pancreatitis: protocol for a systematic review and meta-analysisPLOS ONE

Dear Dr. Xiao,

Thank you for submitting your manuscript to PLOS ONE. After careful consideration, we feel that it has merit but does not fully meet PLOS ONE’s publication criteria as it currently stands. Therefore, we invite you to submit a revised version of the manuscript that addresses the points raised during the review process.

ACADEMIC EDITOR: Please consider reviewer 1's suggestions.

We look forward to receiving your revised manuscript.

Kind regards,

Marcus Tolentino Silva

Academic Editor

PLOS ONE

Journal Requirements:

Reviewers' comments:

Reviewer's Responses to Questions

**Comments to the Author**

1. Does the manuscript provide a valid rationale for the proposed study, with clearly identified and justified research questions?

Reviewer #1: Yes

Reviewer #2: Partly

2. Is the protocol technically sound and planned in a manner that will lead to a meaningful outcome and allow testing the stated hypotheses?

Reviewer #1: Yes

Reviewer #2: Partly

3. Is the methodology feasible and described in sufficient detail to allow the work to be replicable?

Reviewer #1: Yes

Reviewer #2: Yes

4. Have the authors described where all data underlying the findings will be made available when the study is complete?

Reviewer #1: Yes

Reviewer #2: Yes

5. Is the manuscript presented in an intelligible fashion and written in standard English?

Reviewer #1: Yes

Reviewer #2: Yes

6. Review Comments to the Author

You may also provide optional suggestions and comments to authors that they might find helpful in planning their study.

Reviewer #1: This protocol for systematic review aimed to assess the efficacy and safety of Dachaihu Decoction (DCD) for the treatment of acute pancreatitis (AP). I compared the two versions of manuscript and studied the response letter carefully. I think the authors have revised manuscript well, and the current version is almost suitable for publication. I have only two minor comments.

(1) The author decided to include electronic datasets (Scopus, CINAHL), grey literature sources (OpenGrey, British Library Inside, ProQuest Dissertations & Theses Global, and BIOSIS preview) as data sources. Please provide the detailed search strategy for other databases in the Appendix S3. Please also add the preliminary search date and number of retrieved articles in the preliminary search in Appendix S3.

(2) The author declared that this systematic review is necessary because the safety and effectiveness of DCD in AP have not been verified. Please consider to use the PRISMA harms checklist to improve the harms reporting in your systematic review (BMJ. 2016 Feb 1;352:i157. doi: 10.1136/bmj.i157)

Reviewer #2: The authors have answered my comments sufficiently

English is also improved, hence can be accepted.

7. PLOS authors have the option to publish the peer review history of their article (what does this mean?). If published, this will include your full peer review and any attached files.

Reviewer #1: **Yes: **Prof. Weiwu Yao

Reviewer #2: No

---

## [Author Response · Author response to Decision Letter 1]

10 Mar 2023

Dear editor and reviewers:

Thank you very much for your processing and recognition of our manuscript. We have revised the manuscript again according to the opinions of reviewers and editors.

Journal Requirements: Please review your reference list to ensure that it is complete and correct. If you have cited papers that have been retracted, please include the rationale for doing so in the manuscript text, or remove these references and replace them with relevant current references. Any changes to the reference list should be mentioned in the rebuttal letter that accompanies your revised manuscript. If you need to cite a retracted article, indicate the article’s retracted status in the References list and also include a citation and full reference for the retraction notice.

All references have been checked in this revision and we ensure their completeness and accuracy. In addition, we determined that no retracted or amended papers were cited in the manuscript. In this revision, we have added a new reference according to the reviewer's opinion (41. Zorzela L, Loke YK, Ioannidis JP, et al. PRISMA harms checklist: improving harms reporting in systematic reviews. BMJ. 2016 Feb 1;352:i157. doi: 10.1136/bmj.i157. Erratum in: BMJ. 2016;353:i2229. PMID: 26830668).

Reviewer #1: This protocol for systematic review aimed to assess the efficacy and safety of Dachaihu Decoction (DCD) for the treatment of acute pancreatitis (AP). I compared the two versions of manuscript and studied the response letter carefully. I think the authors have revised manuscript well, and the current version is almost suitable for publication. I have only two minor comments.

Thank you very much for your recognition of our work. In addition, for the two comments you put forward in this review, we have revised accordingly and replied one by one below.

(1) The author decided to include electronic datasets (Scopus, CINAHL), grey literature sources (OpenGrey, British Library Inside, ProQuest Dissertations & Theses Global, and BIOSIS preview) as data sources. Please provide the detailed search strategy for other databases in the Appendix S3. Please also add the preliminary search date and number of retrieved articles in the preliminary search in Appendix S3.

We are very grateful for the reviewer's comment, which will help to improve the credibility and repeatability of our research. We have provided detailed search strategies for all databases in Appendix S3, indicating the date of the initial search and the number of documents retrieved. It is worth mentioning that the deadline for our pre-search is February 4, 2023, so we have extended the deadline for the official search in the protocol (Line 39, 183). As described in the manuscript, the official search deadline was set for May 31, 2023.

(2) Line 28-30: The author declared that this systematic review is necessary because the safety and effectiveness of DCD in AP have not been verified. Please consider to use the PRISMA harms checklist to improve the harms reporting in your systematic review (BMJ. 2016 Feb 1;352:i157. doi: 10.1136/bmj.i157).

 We have to admit that this is a very meaningful proposal. We accepted the recommendation and stated in the manuscript that this systematic review and meta-analysis will be performed and reported under the PRISMA harms checklist (Line 144-145).

Reviewer #2: The authors have answered my comments sufficiently. English is also improved, hence can be accepted.

Thank you very much for your recognition. Of course, we are even more grateful for your suggestions in the first review, which is of vital significance to the smooth development of our research.

Yours sincerely,

Xiang Xiao

---

## [Decision Letter · Decision Letter 2]

28 Apr 2023

Efficacy and safety of Dachaihu Decoction for acute pancreatitis: protocol for a systematic review and meta-analysis

PONE-D-21-34892R2

Dear Dr. Xiao,

We’re pleased to inform you that your manuscript has been judged scientifically suitable for publication and will be formally accepted for publication once it meets all outstanding technical requirements.

Kind regards,

Marcus Tolentino Silva

Academic Editor

PLOS ONE

Additional Editor Comments (optional):

Reviewers' comments:

Reviewer's Responses to Questions

**Comments to the Author**

1. Does the manuscript provide a valid rationale for the proposed study, with clearly identified and justified research questions?

Reviewer #1: Yes

Reviewer #2: Yes

2. Is the protocol technically sound and planned in a manner that will lead to a meaningful outcome and allow testing the stated hypotheses?

Reviewer #1: Yes

Reviewer #2: Yes

3. Is the methodology feasible and described in sufficient detail to allow the work to be replicable?

Reviewer #1: Yes

Reviewer #2: Yes

4. Have the authors described where all data underlying the findings will be made available when the study is complete?

Reviewer #1: Yes

Reviewer #2: Yes

5. Is the manuscript presented in an intelligible fashion and written in standard English?

Reviewer #1: Yes

Reviewer #2: Yes

6. Review Comments to the Author

You may also provide optional suggestions and comments to authors that they might find helpful in planning their study.

Reviewer #1: The authors have improve the protocol as I suggested before. I think the protocol is now acceptable for publication.

Reviewer #2: The authors have answered all the concerns that I raised in the original manuscript.

This is an important aspect of the disease and can be accepted

7. PLOS authors have the option to publish the peer review history of their article (what does this mean?). If published, this will include your full peer review and any attached files.

Reviewer #1: **Yes: **Prof. Weiwu Yao, from Tongren Hospital Shanghai Jiao Tong University School of Medicine

Reviewer #2: No

---

## [Editor Report · Acceptance letter]

9 May 2023

PONE-D-21-34892R2 

Efficacy and safety of Dachaihu Decoction for acute pancreatitis: protocol for a systematic review and meta-analysis 

Dear Dr. Xiao:

I'm pleased to inform you that your manuscript has been deemed suitable for publication in PLOS ONE. Congratulations! Your manuscript is now with our production department. 

Kind regards, 

on behalf of

Dr. Marcus Tolentino Silva 

Academic Editor

PLOS ONE